# Reward History Modulates the Processing of Task-Irrelevant Emotional Faces in a Demanding Task

**DOI:** 10.3390/brainsci13060874

**Published:** 2023-05-29

**Authors:** Ning-Xuan Chen, Ping Wei

**Affiliations:** Beijing Key Laboratory of Learning and Cognition and School of Psychology, Capital Normal University, Beijing 100048, China; ningxuanchen@126.com

**Keywords:** reward history, implicit facial processing, emotion, attentional resources depletion, event-related potential

## Abstract

The aim of the current study was to examine how reward-associated emotional facial distractors could capture attentional resources in a demanding visual task using event-related potentials (ERPs). In the learning phase, a high- or low-reward probability was paired with angry, happy, or neutral faces. Then, in the test phase, participants performed a face-irrelevant task with no reward at stake, in which they needed to discriminate the length of two lines presented in the center of the screen while faces that were taken from the learning phase were used as distractors presented in the periphery. The behavioral results revealed no effect of distractor emotional valence since the emotional information was task-irrelevant. The ERP results in the test phase revealed a significant main effect of distractor emotional valence for the parieto-occipital P200 (170–230 ms); the mean amplitudes in both the angry- and happy-face conditions were more positive than the neutral-face condition. Moreover, we found that the high-reward association enhanced both the N170 (140–180 ms) and EPN (260–330 ms) relative to the low-reward association condition. Finally, the N2pc (270–320 ms) also exhibited enhanced neural activity in the high-reward condition compared to the low-reward condition. The absence of emotional effects indicated that task-irrelevant emotional facial stimuli did not impact behavioral or neural responses in this highly demanding task. However, reward-associated information was processed when attention was directed elsewhere, suggesting that the processing of reward-associated information worked more in an automatic way, irrespective of the top-down task demand.

## 1. Introduction

Previous studies have reported that both reward-associated stimuli and emotional stimuli enjoy processing priority [1,2,3,4,5]. Reward learning could guide visual selective attention by optimizing attentional resource allocation to maximize positive outcome acquisition [6,7,8,9,10]. Moreover, reward-associated stimuli could interfere with a primary task when they are task-irrelevant and not rewarded anymore [11,12,13,14,15]. At the same time, emotional faces can capture more attentional resources than neutral ones, in that the former intrinsically imbue greater biological and social significance that may be beneficial to survival [3,16,17,18,19,20,21]. A few studies revealed that the emotional superiority effect exists when emotional content is task-irrelevant [3,22,23]; for example, when participants are asked to discriminate emotional faces along a nonemotional dimension (e.g., facial gender [24] and a contrast-induced line, [25]) or even perform tasks with emotional stimuli as distractors [26,27,28,29]. However, very recent evidence reported that the effect of emotional facial or body expressions is much more context-dependent because this emotional information elicits consistent behavioral effects when it is task-relevant to the current task goal but not when it is task-irrelevant [30,31,32,33]. For example, in a Go/No-go task, participants were required to perform or withhold a movement according to the valence of facial pictures or according to the gender of the same facial pictures. The results revealed that the emotional information affected the motor responses in the former task setting but not the latter case [30].

Combining these two lines of evidence, several recent studies have explored modulated brain responses to emotional stimuli after pairing them with high- or low-reward values, using the event-related potential (ERP) technique. For example, a study by Yao and colleagues [34] paired happy schematic faces with high-reward values and angry schematic faces with low-reward values, revealing an anger-superiority effect, such as, a faster reaction time (RT) and a larger N2pc (N2-posterior-contralateral), which indicates that attentional capture to salient stimuli [35,36] disappeared in a visual search task after these associations were established. Their results showed that the low-reward history associated with angry faces decreased the attention-capture effect by weakening their salience. Moreover, a previous study of ours [37] imbued angry and happy faces with a high or low probability of reward, and in the following test phase, participants performed a visual search task to discriminate the emotion category of the learned face singleton. ERP results revealed a significant interaction between reward history and emotion for the N100 and P200 components, with the difference between the mean amplitudes for angry and happy faces being smaller in the high-reward association condition than the low-reward association condition. This reward-modulation effect indicates that reward association reduces the attentional allocation to angry faces in the perceptual levels of emotional face processing. Together, these results suggest that reward learning can modulate the attentional salience of emotional stimuli at very early stages of processing [38]. A few studies have further revealed that reward-associated neutral faces could attain emotional significance and may share the same processing as emotional faces in the early time window [13,29,39,40,41].

It is worth noting that the emotional faces in these aforementioned studies were task-relevant, which implied that enough attentional resources could be utilized for the processing of both the reward and the emotional information of the critical stimuli. It is intriguing to ask whether the reward learning modulation effect on emotional faces could still exist if the faces were task-irrelevant.

It is widely accepted that how much spare attentional resources remain for the processing of task-irrelevant items depends on the cognitive load of the primary task [26,42,43]. However, rewarded or emotional distractors can still succeed in attentional capture when fewer attentional resources are utilized [13,14,44]. Reward learning could act like ‘emotional tagging’ to facilitate the cognitive processing of stimuli through a bottom-up pattern [45,46]. Exploring the interaction of motivational saliency and the emotional valence of a stimulus in determining attentional processing can help to further reveal how the attentional system can be affected by both motivational and emotional systems [14].

In the current study, we used a similar reward-association procedure during the learning phase as our previous study did [37], but we used a face-irrelevant task in the test phase to examine how reward-associated emotional facial distractors could interfere with an ongoing attention-demanding task. In the learning phase, participants learned to associate angry, happy, or neutral faces with high- or low-reward probabilities. In the test phase, participants were instructed to discriminate the length of two lines in the center of the screen while the reward-associated faces were presented in the periphery as distractors. The staircase method was used to ensure task difficulty and thus, investigate how reward-associated emotional facial distractors could attain attentional capture under an attentional resource depletion condition.

ERPs were recorded during the test phase, with a focus on the emotion-sensitive components, the P200, N170, and EPN (early posterior negativity), and the attentional capture component, the N2pc. The parieto-occipital P200 (170–230 ms) component is sensitive to the onset of pleasant-going arousal-related emotional stimuli [47,48]. The N170 (around 200 ms; temporo-occipital) is sensitive to the configural processing of faces [49,50]. The subsequent EPN (240–340 ms; temporo-occipital) reflects enhanced sensory encoding of emotional faces [51,52] and arousing stimuli [53,54]. Finally, we were interested in a widely used marker for spatial selective attention, the N2pc, which reflects the deployment of attentional resources to the targets or salient distractors at the periphery [34,36,55,56,57].

In accordance with previous studies, we expected slower RTs for the face distractors associated with high reward than the faces associated with low reward [12], but no RT differences between the three kinds of emotional faces [26,27,28]. According to previous ERP results [24,25,37,58], we hypothesized that (1) high-reward-associated faces would elicit more positive-going ERP responses for the P200, N170, EPN, and N2pc components than the low-reward condition would and that (2) emotional facial distractors would elicit stronger brain responses than neutral faces in perhaps earlier time windows. Finally, we were interested in examining whether reward learning could still have a modulating effect on emotional distractors, as revealed by the above-mentioned ERP components.

## 2. Materials and Methods

### 2.1. Participants

Twenty-four undergraduate and graduate students (eleven females and thirteen males; all between 19 and 28 years of age) who were right-handed, had normal or corrected-to-normal vision, and had no neurological or neuropsychological disorder, participated in this study. Our experiment was conducted in accordance with the Declaration of Helsinki and was approved by the Ethics Committee of the Department of Psychology at Capital Normal University (protocol code CNU-20151216). All participants gave informed consent before the experiment. The sample size of 24 subjects was calculated in G-power [59] by setting the partial *η*^2^ to 0.3, α to 0.05, and power (1-β) to 0.95, based on previous related studies [34,60].

### 2.2. Design and Materials

We used a 2 × 3 within-participant factorial design for this experiment. The first factor was the reward possibility (high vs. low), and the second factor was the emotional valence of the distractor face (angry or happy vs. neutral).

Twelve pictures were selected as stimuli from the Chinese Facial Affective Picture System (CFAPS), with both the valence and arousal levels rated on a nine-point Likert scale [61]. There were 4 happy faces, 4 angry faces, and 4 neutral faces, with 2 female and 2 male faces in each category. The facial luminance was controlled by a unified luminance template from Photoshop software. Arousal levels of angry and happy faces were matched (mean [M] ± standard deviation [SD]: happy = 6.80 ± 0.19; angry = 7.33 ± 0.54), and neutral faces significantly differed from happy and angry stimuli in their arousal level (M ± SD: neutral = 4.42 ± 0.174). Moreover, normative valence ratings among the three categories of pictures differed from one another (M ± SD: positive = 6.93 ± 0.32; neutral = 4.45 ± 0.21; negative = 2.57 ± 0.32).

In addition, we created a scrambled image for each corresponding face by slicing, random splicing, and low-pass filtering the original face. Therefore, all the scrambled images shared the visual representation with their corresponding faces. Each scrambled image and the pre-converted target face were presented lateralized in pairs in each trial (see Figure 1).

### 2.3. Procedures

#### 2.3.1. Value Learning Phase

Stimuli were presented by Presentation software (https://www.neurobs.com, accessed on 29 March 2023). Participants were tested in a dimly lit and sound-attenuated room with a fixed viewing distance of 65 cm. Each trial (Figure 1A) began with a white fixation cross (0.4° × 0.4°) at the center of a black screen for 500 ms. Then, two faces (3.52° × 4.47°) were separately presented on the left and right sides of the central fixation cross, with a center-to-center distance between the face and the fixation cross of 2.10°. Six pairs of faces were chosen beforehand, including three pairs of male faces with angry/happy/neutral emotions and three pairs of female faces with the same emotions. Faces in each pair were of the same valence. Upon the presentation of each face pair, participants need to choose one face with the left forefinger by pressing the ‘F’ key on the computer keyboard for the left face and the right forefinger by pressing the ‘J’ key for the right face. Then, feedback was immediately displayed as ‘+25’ or ‘+0’. In each face pair, one face was associated with a high probability (80%) for ‘+25’ feedback and a low probability (20%) for ‘+0’ feedback, and the probabilities were reversed for the other face pair. The total amount of the experimental reward was also displayed under the single trial feedback message. In addition, a sinusoidal sound (500 or 1500 Hz) was played for 500 ms with a message on the screen to remind the participants of the reward value (for similar manipulations, see [37]). The assignment of reward probability and the sound was counterbalanced across participants. Participants were asked to maximize their earnings by choosing the left or right face without being informed about the reward probability distribution.

The value learning phase contained 10 blocks of 60 trials each (a total of 600 trials). Each face pair was displayed ten times in each block. Before the experiment started, all the participants were informed that they would be paid 0.4% of the total experimental money they accrued as bonus cash at the end of the experiment. On average, participants won 45 Chinese Yuan of bonus cash (in addition to the base pay of 40 Chinese Yuan).

#### 2.3.2. Test Phase

During this phase, participants were informed that no reward would be delivered. At the beginning of each trial (Figure 1B), a fixation cross (0.4° × 0.4°) was displayed in the center of the screen for 800 ms. Then, the face, the corresponding scrambled image, and two target lines were presented together with the fixation cross for 150 ms. Faces and corresponding scrambled images were the same size and were presented at the same location as the images displayed during the value learning phase. The two lines were located at 0.75° from the fixation cross, and the length range of the lines was from 1.06° to 1.32°. Participants were instructed to discriminate the length of these lines by pressing corresponding keys on a computer keyboard. The mapping of the response keys was counterbalanced across participants, with ‘F’ (or ‘J’) for identical lengths and ‘J’ (or ‘F’) for different lengths. Participants were instructed to respond as accurately and quickly as possible upon the presentation of the lines. The two bars were of the same length in half of the trials and were of different lengths in the other half of the trials for each emotional or reward association condition. After the fixation was presented for 1500 ms, an empty screen appeared for 1000 ms to 1500 ms before the next trial. The reason for the random inter-trial intervals was to prevent systematic interference from the preceding ERPs [62]. Participants were asked to keep their eyes on the fixation cross to minimize eye movements.

According to each participant’s performance, length differences in the different-length trials were individually modulated in order to keep task difficulty comparable across participants. Here, we used the ‘3-down, 1-up (or 3/1) staircase’ method to manipulate the length difference between the two lines, such that the length difference was decreased by two pixels after every new set of three consecutive correct responses in the different-length trials, and the length difference was increased by two pixels after each error in the different-length trials. The maximum length difference between two lines was 8 pixels and the minimum was 2 pixels. Participants received one practice block with 48 trials before the formal task. The initial length difference was the maximum (e.g., 8 pixels) during this practice phase. The final value of the length difference for each participant during the practice phase was used as that participant’s initial length difference for the first block in the subsequent formal task.

The formal task contained 8 blocks of 48 trials (384 trials in total), with each block having 8 trials for each face in a pseudo-randomized order, and with half same-length trials and half different-length trials for each condition. In a single block, half of the trials showed lines with the same length, while the other half showed lines with different lengths.

### 2.4. EEG Recordings and Analyses

We recorded the electroencephalographic (EEG) data during the test phase using an elastic cap with 62 Ag/AgCl electrodes from the NeuroScan SynAmps system (NeuroScan Inc. Sterling, VA, USA). The electrodes were mounted according to the extended international 10–20 system. The signals were referenced online to the left mastoid and re-referenced offline to the averaged mastoids. Horizontal and vertical electro-oculographic (EOG) recordings were monitored by two additional channels. The EEG and EOG signals were amplified using a 0.01–100 Hz band pass at a sampling rate of 500 Hz. Impedance was kept under 5 kΩ.

The EEG data were preprocessed using the EEGLAB toolbox [63]. Data were band-pass filtered offline from 0.05 to 40 Hz. Eyeblink and movement components were corrected by the Independent Component Analysis (ICA) algorithm [63]. By inspecting the maps of independent components, we picked the independent components of noise and subtracted ICA components from the data to reject the eyeblink and movement noise. The EEG data were epoched from −100 ms to 600 ms relative to stimulus onset. Baseline corrections were performed on the mean activity during pre-stimulus onset. We excluded trials with a voltage of ±75 μV at any electrode. Erroneous trials were also excluded from further analyses. Therefore, the total number of excluded trials was < 30% per condition (94 trials were discarded on average for each participant), leaving over 45 valid trials for each condition for every participant.

Based on the topographical maps of our grand-averaged ERP activity and findings from previous ERP studies using emotional faces [60,64,65], we analyzed the P200, N170, EPN, and N2pc components. We averaged data from parietal-occipital scalp electrodes Pz and POz to index the P200 component between 170 and 230 ms. EEG data were analyzed using repeated-measures analysis of variance (ANOVA) with reward probability (high-reward vs. low-reward association) and emotional faces (angry, happy, or neutral). In addition, the N170 component (140–180 ms) and the EPN component (260–330 ms) were calculated for the lateral temporo-occipital electrodes (P3, P5, PO5, P4, P6, and PO6). A three within-participant factors ANOVA was used on the average amplitudes over the left electrodes (P3, P5, and PO5) and the right electrodes (P4, P6, and PO6) with reward probability (high-reward vs. low-reward association), emotional faces (angry, happy, or neutral), and electrode topography (left vs. right).

Finally, we analyzed the N2pc components that are typically located in the temporo-occipital regions at lateral posterior scalp sites [36,66,67]. Our study measured the PO7 and PO8 electrodes between 270 and 320 ms. The ipsilateral waveform was computed as the average of the electrode ipsilateral to the facial distractor, whereas the contralateral waveform was computed as the average of the electrode contralateral to the facial distractor. Here, we also used ANOVAs with three within-participant factors: reward probability (high-reward vs. low-reward association), emotional faces (angry, happy, or neutral), and contralaterality (ipsilateral vs. contralateral to the facial distractor location).

For all the ANOVAs, the significant alpha level was set to 0.05, and Bonferroni pairwise or simple main effects comparisons were used to supplement the ANOVA results. The Greenhouse–Geisser correction was applied for factors with two or more levels. Note that all the ANOVA results here are reported with corrected *p*-values and uncorrected degrees of freedom.

## 3. Results

### 3.1. Behavioral Results

#### 3.1.1. Learning Task

Based on previous studies [64,65], we calculated the proportion of times that participants chose the high-reward probability faces for each emotional category in each block as the value-learning index. Choice behavior during the learning phase revealed a gradual increase in the proportion of high-reward category selections (Figure 2). An ANOVA was conducted on the choice probability, with blocks (first vs. last block) and the facial emotional valence (angry, happy, vs. neutral) as within-participant factors [64,65]. We found a significant main effect on blocks, *F*(1,46) = 185.72, *p* < 0.001, $\eta_{p}^{2}$ = 0.890. Bonferroni-corrected pairwise comparisons revealed increased high-reward category selection during the last block (97.85%) compared to the first block (60.69%). Neither the main effect of facial emotional valence nor the interaction reached significance; both had a *p*s > 0.05.

#### 3.1.2. Test Phase

The mean RTs and response accuracy in each experimental condition are reported in Table 1. An ANOVA was conducted on the RTs, with reward probability (high vs. low) and the distractor’s emotional valence (angry or happy vs. neutral) as within-participant factors. Neither RTs nor response accuracy revealed any significant effects; all had a *p*s > 0.05.

### 3.2. ERP Results during the Test Phase

Analyses of the P200 component (Figure 3) revealed a significant main effect of distractor emotional valence, *F*(2,46) = 4.217, *p* = 0.021, $\eta_{p}^{2}$ = 0.155. Bonferroni-corrected pairwise comparisons revealed that the mean amplitudes for the angry-face condition (−0.309 μv) were significantly more positive than for the neutral-face condition (−0.849 μv), *p* = 0.05; the amplitudes for the happy-face condition (−0.198 μv) was also significantly more positive than that the neutral-face condition, *p* = 0.022, but there was no difference between the angry-face and the happy-face conditions, *p* > 0.1. Neither the main effect of reward nor the interaction reached significance; both has a *p*s > 0.05.

Analysis of the N170 component (Figure 4) revealed a significant main effect of reward learning, *F*(1,23) = 5.094, *p* = 0.034, $\eta_{p}^{2}$ = 0.181, with the amplitude in the high-reward probability condition (−4.145 μv) significantly less negative than the low-reward probability condition (−4.516 μv). A significant main effect of electrode topography was also found, *F*(1,23) = 14.148, *p* = 0.001, $\eta_{p}^{2}$ = 0.381, with amplitudes from the left electrodes (−3.421 μv) being significantly less negative than the right electrodes (−5.285 μv). No other effects or interactions reached significance; all had a *p*s > 0.1.

Analysis of the EPN component (Figure 4) revealed a significant main effect of reward, *F*(1,23) = 14.603, *p* = 0.001, $\eta_{p}^{2}$ = 0.388, with amplitudes in the high-reward probability condition (0.317 μv) being significantly less negative than the low-reward probability condition (−0.333 μv). No other main effects or interactions reached statistical significance; all had a *p*s > 0.05.

Finally, analysis of the N2pc component (Figure 5) revealed a significant main effect of reward, *F*(1,23) = 7.302, *p* = 0.013, $\eta_{p}^{2}$ = 0.241, with amplitudes in the high-reward probability condition (0.461 μv) being significantly less negative than the low-reward probability condition (−0.047 μv). We also found a significant main effect of contralaterality, *F*(1,23) = 18.520, *p* < 0.001, $\eta_{p}^{2}$ = 0.446, with the mean amplitudes for the contralateral electrode of the distractor faces (−0.278 μv) being more negative than the ipsilateral electrode (0.692 μv). No other main effects or interactions reached significance; all had a *p*s > 0.05.

## 4. Discussion

In the current study, we investigated how reward-associated emotional facial distractors could affect the ongoing task using ERPs. Angry, happy, and neutral facial pictures were paired with a high or low probability of reward during the initial learning phase. Participants then performed a length discrimination task in the center of the screen with the learned face presented at the periphery, and there was no reward provided in the test phase. The behavioral results revealed no effect of reward association or distractor emotional valence in the test phase. The ERP results in this phase revealed a significant main effect of distractor emotional arousal for the parieto-occipital P200 (170–230 ms), in which the mean amplitudes of both angry- and happy-face conditions were more positive than the neutral-face condition. Moreover, both the N170 (140–180 ms) and EPN (260–330 ms) exhibited a significant main effect of reward association, such that the mean amplitudes of both components in the high-reward conditions were significantly more positive than the low-reward condition. Finally, we also observed a significant main effect of reward association on the N2pc (270–320 ms), with the mean amplitudes in high-reward conditions being significantly more positive than low-reward conditions.

For the parieto-occipital P200 (170–230 ms), we found a main effect of emotion, with both angry and happy faces eliciting more positive amplitudes than the neutral faces elicited. The parieto-occipital P200 effect is assumed to indicate the processing of emotional intensity or arousal [47,68,69,70], with this effect often being enhanced for both negative emotional stimuli [71,72] and positive emotional stimuli [21]. Although the later attentional component (e.g., N2pc) has been found to be modulated by the task relevance of peripheral emotional stimuli, the early component was often found to be modulated by the mere presence of the emotional stimuli even in the periphery [25,58]. In our study, the P200 showed more positive amplitudes for both emotional (angry/happy) faces than nonemotional (neutral) faces, irrespective of the valance of the faces. Since the arousal levels of angry and happy faces were matched and significantly differed from the neutral faces, and the valence levels among the three categories of faces differed from one another; we interpreted the effect in P200 as reflecting the effect of arousal. Moreover, the current length discrimination task was manipulated to be fairly demanding so there would be fewer spared attentional resources left; hence, the results suggested that the emotional facial distractors could merely be discriminated as emotional/nonemotional stimuli along the arousal dimension in the early time windows. Meanwhile, no reward effect was found in the P200 components, which was inconsistent with our previous study that found a Reward × Emotion interaction for the P200 in a visual search task, in which participants were asked to discriminate the emotional valence of a face singleton previously associated with reward [37]. Taking these two studies together, these results indicate that the sensitivity to motivational significance on the P200 was modulated by available attentional resources and was affected by task relevance.

The N170 component exhibited a main effect of the reward, such that the mean amplitudes in high-reward conditions were significantly more positive than the low-reward conditions. N170 has been widely accepted to reflect facial or emotional processing [73,74,75,76]. However, we found no emotional effects on the N170 in the current results, which is consistent with the view that the N170 is not sensitive to implicit facial emotional processing [17,64,77,78,79,80,81,82]. For example, Eimer and colleagues [78] used a length discrimination task with bilateral facial distractors and found that the N170 component was not affected by emotional faces. A recent study by Burra and Kerzel [82] also presented faces in the periphery and reported that N170 amplitudes were modulated by emotional valence in a face-relevant task (i.e., identifying the facial gender), but this effect was not present in a face-irrelevance task (i.e., identifying a pixel in the fixation with emotional faces as distractors). These studies suggest that facial structural encoding (reflected from the N170) is insensitive to implicit emotional processing, which is in line with the appraisal theories of emotions [83] but not the motivational theories of emotions [84]. In addition, the demanding task at hand might also result in the absence of an emotional effect on the N170, which is consistent with a study by Müller-Bardorff and colleagues [29], in which the valance effect of facial distractors on the N170 disappeared in the high-load condition compared with the low-load condition.

Furthermore, a few recent studies have consistently reported that reward-associated stimuli could modulate this component, suggesting that the N170 may be sensitive to motivationally significant stimuli in a broader way [24,85,86]. Although attention was directed away from emotional faces in the current study, high-reward associated faces still showed stronger processing priority than low-reward faces, while emotional expression effects were not observed. This implies that reward history may have a stronger superiority effect than intrinsic emotional facial characteristics (e.g., physical structure, valence) even under a high perceptual load.

Moreover, the EPN component exhibited a reward effect similar to the N170, with more positive deflection for the high-reward conditions than for the low-reward conditions. At the same time, the EPN did not exhibit any emotional effects, which is inconsistent with the view that the EPN represents attentional orientation toward emotional stimuli in an automatic way regardless of task demand [87,88], but again is consistent with the view that the modulation of emotional distractor faces is task-dependent [30,31,32]. The EPN may be sensitive to emotional information only when explicit attention is allocated to the emotional dimension [24,89,90,91,92]. For example, Wu and colleagues [24] used two modified MID (Monetary Incentive Delay) tasks, in which participants were required to discriminate the gender of the target face or discriminate whether a number superimposed on a face was even or odd, so the emotional information in both tasks was task-irrelevant. The authors reported that the EPN showed the effect of reward but no effect of emotion in both tasks, which is consistent with the pattern of the current results. These findings indicate that reward learning could attain processing in a more intuitive way, even under the condition of attentional resource depletion, but the processing of emotional information is modulated by task relevance.

Interestingly, although the difference between the contralateral and ipsilateral N2pc waveforms was neither modulated by reward association nor emotional valence, a main effect of reward was observed for the N2pc component, with the high-reward condition eliciting a more positive brain wave than the low-reward condition. The N2pc component is assumed to reflect spatial selective attention, and the larger N2pc difference wave indicates a stronger attentional capture effect [34,36]. In a recent experiment by our group, in which participants were asked to judge the emotion or gender of a reward-associated emotional face presented at the left or right side of the visual field, the N2pc difference wave showed an aligned pattern with the behavioral reaction times, with a larger N2pc associated with shorter reaction times [93]. This suggests that the N2pc reflects the strength of attentional selection when peripheral emotional faces are task-relevant. However, this pattern disappeared when the peripheral face was task-irrelevant, as in the current study, which is in line with recent findings that the emotional effect on the N2pc was modulated by the task relevance of the emotional information [25,58]. It has been suggested that the activation of the frontoparietal attention network needs the integration of both perceptual salient and task relevance of facial stimuli [25,94,95]. As the emotional faces were task-irrelevant and had less motivational value in this experiment, they could not activate the frontoparietal attention network, and this may explain the absence of emotional and reward effects on the N2pc difference wave. However, the positive-going ERPs for both the contralateral and the ipsilateral waveforms in the high-reward condition relative to the low-reward condition might reflect general arousal caused by reward learning, but not an attentional direction to either peripheral field.

## 5. Conclusions

We asked participants to discriminate the length of the target bars in the current study while presenting reward-associated emotional faces at the periphery. The results revealed that the early P200 showed the modulation of both positive and negative emotional faces compared to neutral faces. It implied that the emotional facial distractors could merely be discriminated as emotional/nonemotional stimuli along the arousal rather than the valence dimension. In addition, high-reward associated distractors elicited more positive ERP responses for originally assumed emotion-relevant components (i.e., N170 and EPN). This indicated that reward-associated task-irrelevant stimuli may receive processing even when a highly demanding task is required. We also found a similar effect of reward learning on the attentional orienting component (N2pc), which further implied the stronger attentional deployment to high-reward-associated information. The reward-associated effect on N170, EPN, and N2pc works more in an automatic way despite the top-down task demand. However, there were no emotional effects in these time windows and behavioral results, indicating that task-irrelevant emotional facial stimuli receive no processing when attention is directed elsewhere in this highly demanding task.

## Figures and Tables

**Figure 1 brainsci-13-00874-f001:**
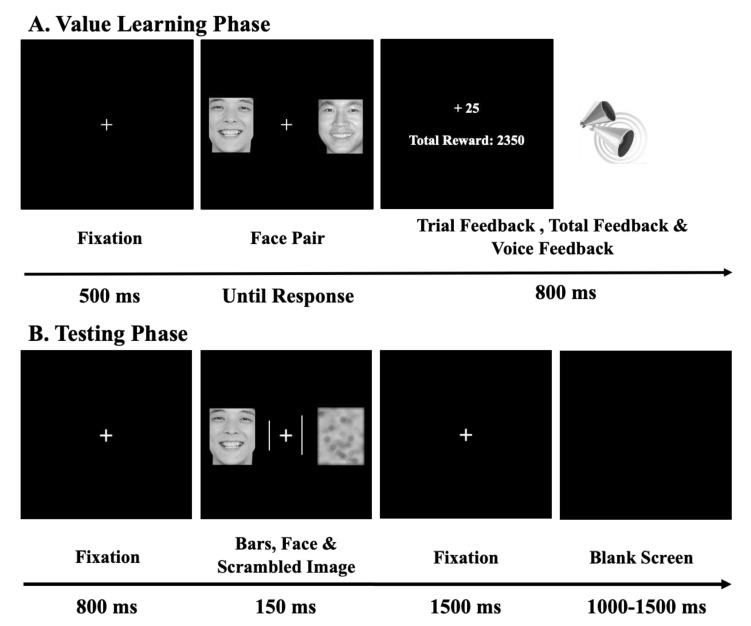
Trial sequence. (**A**) An example of the trial sequence in the value-learning phase. The task in this phase was to choose the left or right face to maximize earnings on each trial. (**B**) An example of the trial sequence in the testing phase. Participants had to discriminate whether the two bars had an equal length.

**Figure 2 brainsci-13-00874-f002:**
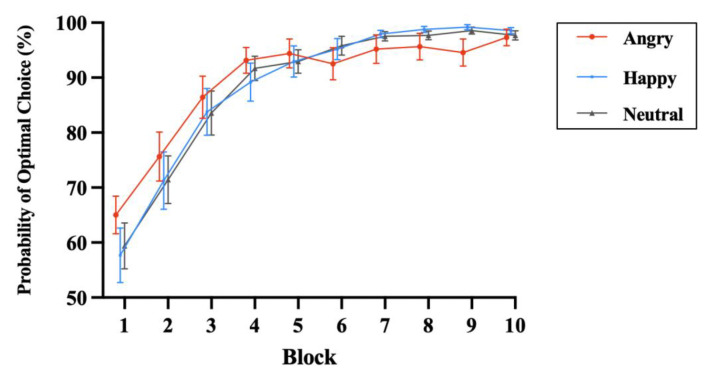
The average probability of choosing the optimal face across participants for each emotional category in each block. Error bars denote the standard error of the mean.

**Figure 3 brainsci-13-00874-f003:**
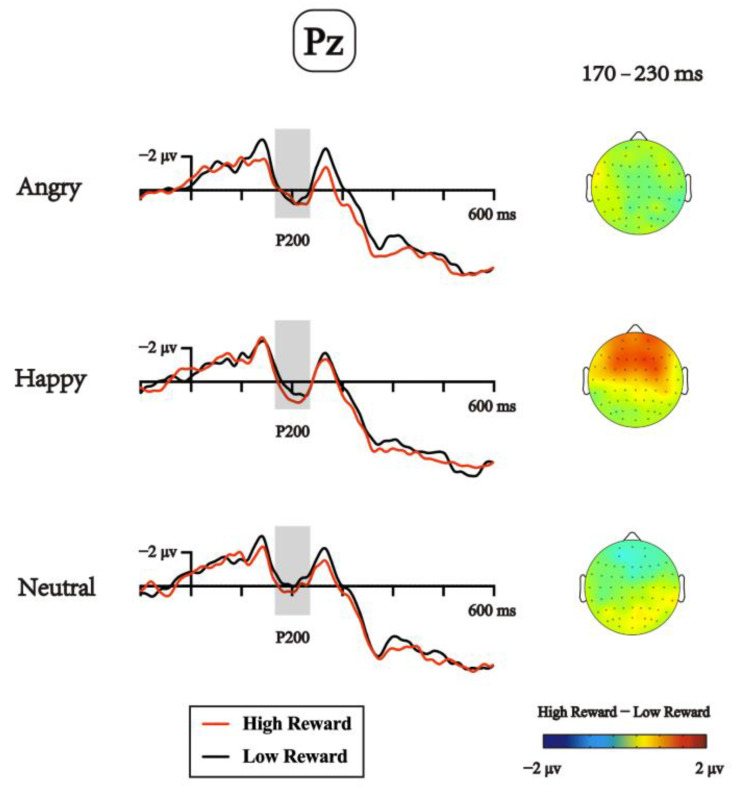
Grand average waveforms at the Pz electrode showing the potentials produced in response to the presentation of the facial distractor in the experiment. The grey shaded areas indicate the time windows for P200, with the mean amplitudes being more positive for both the angry and happy conditions than the neutral condition.

**Figure 4 brainsci-13-00874-f004:**
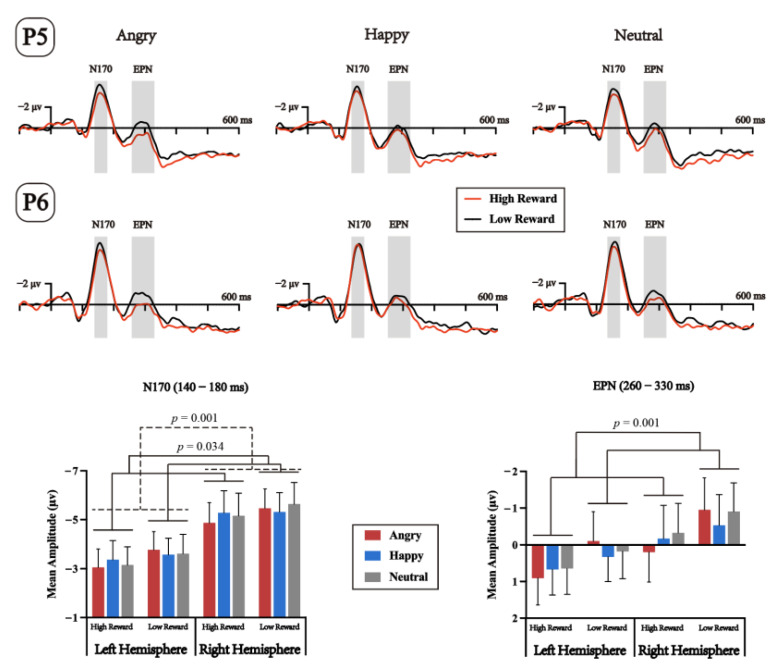
Grand average waveforms at the P5 (left hemisphere) and P6 (right hemisphere) electrodes showing the potentials produced in response to the presentation of the facial distractors in the experiment. Positive voltage is plotted downwards. The grey shaded areas indicate the time windows for N170 and EPN. Bars of the mean amplitudes of N170 and EPN are shown below. Error bars denote the standard error of the mean.

**Figure 5 brainsci-13-00874-f005:**
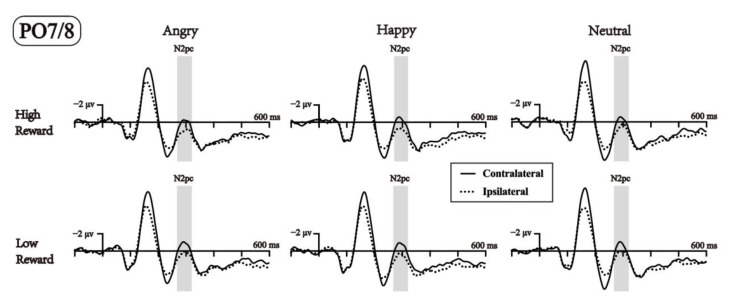
ERPs elicited at PO7 and PO8 in response to the facial distractor are shown separately for trials where the face appears contralateral (solid lines) and ipsilateral (dash lines) to the recording electrode. The grey shaded areas indicate the time windows for N2pc.

**Table 1 brainsci-13-00874-t001:** Mean RTs and accuracy rates in the experimental conditions.

	High-Reward			Low-Reward		
	Angry	Happy	Neutral	Angry	Happy	Neutral
RTs (SE)	715 (16)	721 (17)	716 (18)	717 (18)	717 (17)	721 (17)
Accuracy (SE)	77.6 (1.2)	79.5 (1.2)	76.5 (1.3)	77.4 (1.2)	78.7 (1.3)	80.1 (1.1)

Note: RTs are in ms; accuracy rates are percentages; SE = standard error of the mean.

## Data Availability

The data presented in this study are available upon request from the corresponding author.

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
