# Peer review of "Reward History Modulates the Processing of Task-Irrelevant Emotional Faces in a Demanding Task"

_brainsci, 2023, doi:10.3390/brainsci13060874_

Round 1
Reviewer 1 Report
The authors addressed a timely, engaging, and debated topic as they investigated the effect of reward-associated emotional facial distractors on behavioral performance and the ERPs in a task where 23 participants had to discriminate the length of two lines while presenting reward-associated emotional faces at the periphery. The authors found no effect of such stimuli on measured behavioral parameters (reaction times, RT, and accuracy). However, the four ERP peaks on which the authors focus had an effect. The P2 component was larger for happy and angry faces than for neutral ones, irrespective of reward association. The N170, EPN, and N2pc were larger for high- than for low-reward associated distractors but did not show any effect of emotional valence.
The paper is well written, the methods are sound, and the statistical analysis is well done. However, how the authors introduced and later interpreted, the data needs to be corrected. I trust their results, but they have to change the theoretical frame. As I will illustrate in the following, their data are fully compatible with the novel view according to which facial (or body posture) emotional expressions affect behavioral reactions only when task-relevant. Thus I will ask for major revisions.
MAJOR POINTS
1. The authors support that emotional stimuli automatically elicit behavioral reactions in the observer, in line with the motivational theories of emotions (Bradley, Codispoti et al. 2001). However, a series of very recent studies showed that, differently from what has been thought for a long time, emotional facial or body expressions elicit a consistent, replicable behavioral effect only when they are relevant to participants' goals, i.e., the effect of emotions is context-dependent (Mancini et al. 2020, Mirabella et al. 2022 but there also more works). Mancini et al., (2020) using two versions of a Go/No-go task, showed that emotional facial expressions (happy, angry and fearful) affect motor readiness and accuracy of reaching arm movements only when they are task-relevant. Threatening expressions increased the reaction times (RTs) and the percentage of omission errors, i.e., instances in which participants did not move although they had to, with respect to happy faces, provided that the stimuli' emotional contents were needed to give the correct response. Differently, when the same images were shown, but participants had to move or withhold their movements according to the faces' gender, differences between happy and threatening faces disappeared. Mirabella et al. (2022) showed that whole-body movements (forward gait initiation, GI) share the same features as reaching arm movements regarding emotional stimuli, only when their conscious appraisal was requested. Such finding is relevant as whole-body movements represent a more ecologically valid model as moving the whole body toward or away from an emotional stimulus decreases or increases the physical distance between the stimulus and the self. Finally the same effect has been shown on inhibitory control (Mancini et al. 2022). Overall these studies sustain the appraisal theories of emotions (Scherer and Moors 2019). The authors' results are perfectly in line with such an interpretation. First, the authors found no effect of facial expression in the task where faces were irrelevant. Clearly, this is incompatible with the idea that 'The emotional superiority effect also exists when emotional content is task-irrelevant' (lines 34-5). However, it is fully compatible with the crucial importance of task-relevance. Second, the effect on the ERP components never shows an effect of emotional valence. In fact, the impact on the P2 component is not explained by the valence of the expressions (happy and fearful faces have opposite valences, but they evoke the same effect) but by the different arousal of emotional versus neutral faces (Lines 122-124). No other effects of emotional stimuli were found in ERPs components. By contrast, the association of facial expressions with the reward had a large effect. The authors should discuss these findings to provide a better interpretation of their findings.
2. DISCUSSION. Line 306-7; 317-318 The effect is due to the arousal of the facial images and not to the valence.
MINOR POINTS
1.Line 110 Please provide the protocol number of the Ethics committee.
2.Line 208 How many trials were discarded on average?
References
Bradley et al (2001). "Emotion and motivation I: defensive and appetitive reactions in picture processing." Emotion 1(3): 276-298.
Mancini et al (2020). "Threatening Facial Expressions Impact Goal-Directed Actions Only if Task-Relevant." Brain Sci 10(11).
Mancini et al. (2022). "Happy facial expressions impair inhibitory control with respect to fearful facial expressions but only when task-relevant." Emotion 22(1): 142-152.
Mirabella et al (2022). "Angry and happy expressions affect forward gait initiation only when task relevant." Emotion.
Scherer & Moors (2019). "The Emotion Process: Event Appraisal and Component Differentiation." Annu Rev Psychol 70: 719-745.
Reviewer 2 Report
In this paper, Chen & Wei investigated if reward history modulates the processing of task-irrelevant emotional Faces in a decision-making task. The paper is written in an easy-to-follow manner but I have a few concerns about the methods, results and the conclusion.
1. First, how do the authors know that their findings were associated with the reward association of faces in the learning period or just simply the inherent emotional valance of the faces? A double disassociation is necessary to clarify the claim.
2. Was the similar vs dissimilar bar length conditions balanced with different facial valences and reward associations? I mean for example, did the similar length condition always occur with a particular facial expression or was this random as well as balanced?
3. Line 148 - “choose” - how do the participants make a choice? By making a saccade, pressing a button? This was not clear and should be mentioned.
4. Why wasn’t the test phase rewarded? Will that impact task performance?
5. Were the participants required to maintain fixation throughout the experiment? No information about this or eye tracking or exclusion criteria with respect to failure of fixations is provided.
6. The authors look at the EEG filtered from 0-40 Hz. This does not include most of the Gamma band (30-70 Hz) (Ray 2011 10.1371/journal.pbio.1000610; Chalk 2010 10.1016/j.neuron.2010.03.013) as well as high gamma (80-150 Hz) (Ray et al., 2008 https://doi.org/10.1016/j.clinph.2007.09.136) which is predominantly associated with attention. On the other hand, 13-30 Hz which is the majority of what the authors considered in this paper is beta band which mostly captures the voluntary goal directed movement preparation and execution (Sendhilnathan et al., 2017 https://doi.org/10.1073/pnas.1703809114 and Sendhilnathan et al., 2021 https://doi.org/10.1073/pnas.2006372118). With this said, I am not sure how the authors support their claim of studying attention using frequencies that do not represent or correlate with attention. This needs to be clarified and discussed - if required, a re-analysis should also be performed.
Minor - please change possibility to probability in line 116
The language is fine.
Reviewer 3 Report
The paper is related to the search for neurocognitive components of EP in the perception of faces with different emotions and activation of attention. The work is difficult for the unprepared reader. The content is understandable only to EEG-EP neurophysiologists. In this regard, in order to improve the work, I recommend the authors to pay attention to the following features:
1. The theme comes from projective techniques that have been used in recent decades in applied psychology, psychodiagnostics and psychotherapy. The authors of this study immediately associate their work with EP waves, which narrows the topic very much. For the Introduction and Discussion, it makes sense to turn to well-known papers (Szondi L., etc.) in order to lead the reader to an understanding of the EP waves.
2. The authors use different symbols to designate EP components. Although the designations of the EP have long been invented - polarity (N, P) and latency (msec). In the main text and in the figures, it is necessary to unify the designations of the EP components. Leave the author's features in the Methods Chapter.
3. Fig. 2. - not very clear. It is necessary to decipher 10 blocks. This information is in the Methods, but is not associated with this figure.
4. In fig. 3-5 need indicate significant differences where possible. There are differences in the text, but differences should be indicated in the figures. You can't see any difference in the pictures at all. It has to do with the amplifier settings. You could choose other settings for clarity in future.
5. Conclusion build strictly on the components. There are 3-4 components in the analysis, which should be brought to the Conclusion. Even if there are no significant differences, this is also the result.
Round 2
Reviewer 1 Report
After the first round of review, the authors were very responsive to my criticisms and improved the manuscript significantly. However, I have a few more points I like the authors will address in the next version.
- ABSTRACT and CONCLUSION I suggest reporting in both sections that, when task-irrelevant, the emotional stimuli do not impact behavioral and neural responses.
- As on line 39 ' body expressions' are mentioned, I suggest citing this paper (Calbi et al. (2022). "Emotional body postures affect inhibitory control only when task-relevant." Front Psychol 13: 1035328"), which deals explicitly with this issue.
- Lines 41-2 I strongly suggest rephrasing the sentence: 'The emotional stimuli affected motor responses only when participants had to move at the presentation of valenced stimuli and to inhibit the response whenever neutral stimuli were presented (emotional discrimination task). By contrast, when the same pictures were shown, and participants had to respond according to the actors' gender, disregarding the valence of the stimuli (in the Gender discrimination task), emotional stimuli did not elicit any behavioral effects". This is because there is no 'superiority effect' in the cited experiments but a response modulation as a function of the stimuli' valence.
- Line 99 The sentence is incorrect. Lease clarify
- Line 153 Change to 'Figure 1A'
- Line 285 Add 'during the Test Phase'
- Line 338. Here add also the fact that no behavioral effects were found in the test phase
- Line 414 delete 'wave', which probably was left from previous versions.
- Lines 448-449 Please clarify the meaning of this sentence.
Reviewer 3 Report
The article looks good in its current form. But in my opinion the conclusion spoils the general perception. The results focus on 4 components of evoked potentials, which are also discussed in depth. Perhaps it will look better if, in the conclusion, for each component, make several lines separately. For this, everything is in the Discussion and will not take much time.
